# A Primal-Dual Formulation for Deep Learning with Constraints

**Yatin Nandwani, Abhishek Pathak, Mausam and Parag Singla**
Department of Computer Science and Engineering
Indian Institute of Technology Delhi
{yatin.nandwani,abhishek.pathak.cs115,mausam,parags}@cse.iitd.ac.in

## Abstract

For several problems of interest, there are natural constraints which exist over the output label space. For example, for the joint task of NER and POS labeling, these constraints might specify that the NER label 'organization' is consistent only with the POS labels 'noun' and 'preposition'. These constraints can be a great way of injecting prior knowledge into a deep learning model, thereby improving overall performance. In this paper, we present a constrained optimization formulation for training a deep network with a given set of hard constraints on output labels. Our novel approach first converts the label constraints into soft logic constraints over probability distributions outputted by the network. It then converts the constrained optimization problem into an alternating min-max optimization with Lagrangian variables defined for each constraint. Since the constraints are independent of the target labels, our framework easily generalizes to semi-supervised setting. We experiment on the tasks of Semantic Role Labeling (SRL), Named Entity Recognition (NER) tagging, and fine-grained entity typing and show that our constraints not only significantly reduce the number of constraint violations, but can also result in state-of-the-art performance.

## 1   Introduction

Deep neural models have become the state of the art in many domains including vision, NLP and speech processing. In the vanilla setting, they are trained end to end from data and without additional knowledge about the task (other than neural architecture and loss function). However, for many problems of interest (e.g., structured prediction or multi-task learning), there is a set of natural constraints which need to be satisfied over the output variables. For example, for the task of NER and POS labeling, the constraint might specify that a word which is given the NER label 'institution' must have the POS label 'noun' or a 'preposition'. Or in 3D human pose estimation from a single view, one may impose symmetry constraints, like equal length of two arms, equal distance of shoulders from the spine etc. (Márquez-Neila *et al.* [2017]). These constraints can be seen as additional background knowledge made available by the domain experts. Incorporating these constraints into a model can presumably regularize the output space resulting in improved predictions.

One line of work trains the neural models without this knowledge, but imposes constraints at inference time (e.g., Lee *et al.* [2019]). We argue in this paper that this is bound to be sub-optimal since the original training of the network was done oblivious to the constraints. Though the deep network, in principle, could learn these directly from the data, but, in practice, this is true only when the available training data is large. Our experiments reveal a large number of constraint violations from such unconstrained models, when trained in low data settings. Rather, modeling the constraints explicitly during training gives a strong prior to the model – it not only reduces constraint violations but can also result in significantly improved predictions by making the training *constraint-aware*.

In this paper, we present a principled solution to the problem of learning a deep network with a given set of hard constraints on output labels. We first formulate this as a constraint optimization problem wherein we maximize the original learning objective (e.g., cross entropy) subject to the constraints being satisfied for each example in the training data.When the network makes predictions in form of probabilities over the output variables, we can rewrite the constraints on output labels as constraints on probabilities output by the network using soft logic (Bröcheler *et al.* [2010], Novák [1987]). This rewrite can be seen as imposing constraints over the set of allowed distributions over the output space, i.e., allowing only those distributions that satisfy the constraints. We then convert this problem into a Lagrangian formulation, with one Lagrange variable per constraint. We solve it using alternating min-max based optimization. Though the resulting problem can be highly non-convex non-concave, convergence guarantees to a local min-max point (in the limit) follow from the theory of min-max optimization (Jin *et al.* [2019]). Since our constraints are specified over the predicted variables (no target values are involved), our formulation easily extends to semi-supervised setting where the unlabeled data only contributes to constraint terms in the formulation.

We note that there have been a few recent attempts at adding constraints during training time, but with significant differences from our work. These include the work by Xu *et al.* [2018], Mehta *et al.* [2018] and Diligenti *et al.* [2017b]. While these existing approaches model the constraints as soft and incorporate a constraint violation penalty directly in the loss term, our constraints are modeled as hard, and we resort to a full Langrangian based optimization. The existing work can require an exponential sum to be computed (Xu *et al.* [2018]), require an additional constraint violation penalty as input with no explicit convergence guarantees (Mehta *et al.* [2018]), or require a specific functional form for the constraints (Diligenti *et al.* [2017b]). In contrast, our formulation is tractable because we do not compute an exponential sum, requires no additional constraint violation term, converges to the stationary point of the objective function, and does not assume a specific functional form for the constraints. We detail these differences in the related work.

We experiment on three different NLP tasks: (a) semantic role labeling, (b) NER tagging, and (c) fine grained entity typing over hierarchical label space. Our experiments clearly demonstrate that, in low data setting, our constraint based learning not only reduces the number of violations, but also result in significantly improved prediction accuracy compared to unconstrained baselines as well as vanilla post processing of constraints at inference time. Semi-supervised learning results in further improvements using our formulation. Furthermore, in some cases, our approach altogether eliminates the need for post-processing of constraints, since they have already been learned by the neural model. For two of the tasks, we obtain state-of-the-art results for small data sizes. On NER, our constrained learning completely eliminates the need for further post-processing with constraints, saving on precious inference time.

Our contribution in this paper can be summarized as follows: (1) We present a principled approach for incorporating domain knowledge in the form of hard constraints. We present a Lagrangian based formulation for learning with constraints in a deep network. Our constraints make use of soft rules to deal with logical operators. (2) We employ a min-max based optimization to solve our constrained formulation. To the best our knowledge, we are the first to use a min-max based formulation for learning with constraints specified over output variables in deep networks. Convergence to local min-max points (Jin *et al.* [2019]) in the limit follows from the theory of min-max optimization. (3) We present experimental results on three different tasks demonstrating the effectiveness of our approach, while achieving state-of-the-art results in two of the domains and also significantly reducing the number of constraint violations in each case.

## 2 Related Work

Use of hard (or soft) constraints in machine learning models predates modern deep learning. Much of this work is concerned with constrained inference. Constrained conditional models use integer linear programming to perform inference with global hard constraints (Roth and Yih [2005], Chang *et al.* [2013]). Other approaches have used dual decomposition to solve locally decomposable constraints (Rush and Collins [2012]). A recent attempt incorporates constraints during inference by incrementally adjusting the learned weights of the network forcing the probability of currently predicted non-satisfying state towards zero (Lee *et al.* [2019]).

Our focus in this work is learning with constraints. Posterior regularization (Ganchev *et al.* [2010]) adapts the learned distribution (post facto) so as to satisfy structural constraints over latent variables

in expectation. Chen and Deng [2013] have employed a primal-dual based formulation for optimizing with constraints in deep models, but their constraints are specified over the weights in a recurrent neural network and are only concerned with imparting stability to the overall learning algorithm. In deep learning models, one of the ways to regularize the output space is through a CRF layer (Koller and Friedman [2009]) at the end of a deep network (Lample *et al.* [2016]). This has met with partial success in vision (Knöbelreiter *et al.* [2017]) as well as in NLP with some state-of-the-art models deploying this either as a post processing step or jointly integrated with training (Huang *et al.* [2015], Chen *et al.* [2018]).

There have been some recent attempts to explicitly incorporate constraints over the output space during training of a deep network. Hu *et al.* [2016] perform posterior regularization over the weights being learned at each step, so that the resultant distribution satisfies a given set of logical rules (constraints). This rule regularized network (teacher) is then used to guide the learning of the original network (student) which balances between optimizing the likelihood based objective and mimicking the teacher network. Their work is different from ours in two main aspects. (1) The imposition of constraints in their algorithm is only indirect by mimicking the rule regularized network. In contrast, we optimize for satisfying the constraints directly. (2) They model constraints as soft whereas we deal with hard constraints. Further, they could achieve limited success in their experiments.

Xu *et al.* [2018] incorporate constraints by forcing the probability of states violating the constraints to zero. They model this as a soft constraint by incorporating a constraint violation penalty in the loss function. More importantly, they need to (pre-)compile a circuit for every constraint in order to compute the sum over all the non-satisfying states. Computing these circuits can be NP hard in many cases leading to intractability. In contrast, we model constraints as hard through the use of Lagrangian. Our formulation disallows the non-satisfying states directly through the use of constraints, and does not require an exponential sum to be computed.

Mehta *et al.* [2018] present an approach for learning with constraints and demonstrate the effectiveness of their method for the specific task of Semantic Role Labeling (SRL). They model the constraints by adding another term to the loss which penalizes the currently predicted state for violating the constraint. They require an additional constraint violation penalty from the designer. Their approach can be seen as a local search in the weight space, so that the resultant weights result in a satisfying assignment. They do not have a global metric which optimizes the weights for satisfying the constraints (e.g, forcing the non-satisfying states to zero probability), and it not clear if their algorithm will converge in the limit. In contrast, we can provide convergence guarantees to min-max points of the objective and we experiment on a variety of NLP tasks.

Diligenti *et al.* [2017b] propose an approach for learning with constraints, where the constraints are specified as logical formulas. Though their approach seems similar to ours at the outset, there are some important differences. Unlike us, they do not work with full Lagrangian formulation. Their approach is simply modifying the loss, and can not handle hard constraints. Further, they require the constraint function to be of specific form (i.e., functionals in the range $[0, 1]$), and present experiments only on a single task. Our work makes no such assumption about the form of constraints and we experiment on a variety of tasks.

## 3 Constrained Learning of Neural Models

Consider learning a neural network over a set of training examples given as $\{x^{(i)}, y^{(i)}\}_{i=1}^{m}$. Each $x^{(i)} \in \mathcal{R}^n$ represents an $n$-dimensional feature vector in a real valued space. Each $y^{(i)} \in \mathcal{V}^r$ represents an $r$ dimensional target (label) vector, where each element of the vector takes values from a discrete (or continuous) valued space denoted by $\mathcal{V}$. Note that $r$ may be input-dependent, e.g., in sequence labeling tasks. Given the parameters $w$ of the network, let $l_w(\hat{y}^{(i)}, y^{(i)})$ denote the loss obtained by predicting $\hat{y}^{(i)}$ when the target is $y^{(i)}$. For instance, this could be the cross entropy loss when the labels are discrete. The goal of learning is to find a set of parameters $w^*$ of the network such that the average loss $L(w) = \frac{1}{m} \sum_{i=1}^{m} l_w(\hat{y}^{(i)}, y^{(i)})$ between the network outputs $\hat{y}^{(i)}$ and the target values $y^{(i)}$ is minimized, i.e., $w^* = \operatorname{argmin}_w L(w)$.

In this work, we are interested in a scenario where we are additionally provided with a set of (hard) constraints which hold over the output label space. We assume that these constraints are provided to us by the domain experts and are available in the form of background knowledge. Our goal is to incorporate this background knowledge to learn a more robust and generalizable model. Our

formulation is based on constructing a Lagrangian, which tries to minimize the original objective subject to the given constraints. We solve our problem using an alternating optimization over a max-min formulation.

## 3.1 A Lagrangian-based Formulation

Let us assume we are given a set of $K$ constraints as $\{C_1(\hat{y}), C_2(\hat{y}), \cdots, C_K(\hat{y})\}$. We will use index $k$ to vary over the constraint set. Each constraint is a function of the predicted values $\hat{y}$ on a given example $x$. Since each of the network outputs in turn is directly a function of weights $w$ of the network (for a given $x$), for ease of notation, we will simply write the constraint set to be $\{C_1(w), C_2(w), \cdots, C_K(w)\}$. Note that the dependence on input vector $x$ is implicit in this notation. Further, without loss of generality, we will assume that each of our constraints $C_k(w)$ is expressed as an inequality constraint over an appropriately defined function $f_k(w)$, i.e., $C_k(w) : \{f_k(w) \leq 0\}$. This can also model constraints of the form $\overline{f}_k(w) = 0$ by replacing them with two inequality constraints of the form $\overline{f}_k(w) \leq 0$ and $-\overline{f}_k(w) \leq 0$.

When dealing with constraints over the set of $m$ training examples, we will incorporate the dependence on the $i^{th}$ example by including the index in the super script, i.e., we will denote the $j^{th}$ constraint over the $i^{th}$ example as $C_j^i(w)$. We are now ready to define our constrained formulation.

$$\text{argmin}_w L(w) \text{ subject to } \quad f_k^i(w) \leq 0; \ \ \forall 1 \leq i \leq m; \ \ \forall 1 \leq k \leq K. \tag{1}$$

One problem with the above formulation is that it has $O(mK)$ number of constraints. In particular, the number of constraints grows linearly with number of examples, which may become unwieldy. We use the following trick to reduce the number of constraints. Since we are only interested in eliminating the states that do not satisfy the constraints, we can in fact ignore the value the function $f_k^i(w)$ takes when the corresponding constraint is satisfied. Accordingly, we define the Hinge function $H : \mathcal{R} \to \mathcal{R}$ as: $H(c) = c \ \ for \ \ c \geq 0$, and $0 \ \ for \ \ c < 0$.

We equivalently replace each constraint of the form $f_k^i(w) \leq 0$ by $H(f_k^i(w)) = 0$ without changing the original formulation. Intuitively, $H(f_k^i(w))$ can be thought of as describing the loss incurred when the corresponding constraint is not satisfied. This loss is zero when the constraint is satisfied. This transformation will be useful in the next step when we combine together instances of a single type of constraint applied to different examples in the training set. In the new formulation, we get our primal objective as:

$$\text{argmin}_w L(w) \text{ subject to } \quad H(f_k^i(w)) = 0; \ \ \forall 1 \leq i \leq m; \ \ \forall 1 \leq k \leq K. \tag{2}$$

Clearly, $H(f_k^i(w)) \geq 0, \forall i, k$ by definition. Therefore, a necessary and sufficient condition to enforce $\forall i : H(f_k^i(w)) = 0$ is $\sum_i H(f_k^i(w)) = 0$. This is true for all $k$. Defining $h_k(w) = \sum_i H(f_k^i(w))$, we can therefore write our primal objective in Equation 1 as:

$$\text{argmin}_w L(w) \text{ subject to } \quad h_k(w) = 0; \ \ \forall 1 \leq k \leq K. \tag{3}$$

A standard way of solving the optimization problem described in Equation 3 is to find a stationary point of the corresponding Lagrangian, $\mathcal{L}$

$$\mathcal{L}(w; \Lambda) = L(w) + \sum_{k=1}^{K} \lambda_k h_k(w) \tag{4}$$

Here, $\Lambda = \{\lambda_k\}_{k=1}^{K}$ denotes the $K$ sized vector of Lagrange multipliers. Note that since $h_k(w)$ is always non-negative, the constraint $h_k(w) = 0$ is equivalent to $h_k(w) \leq 0$. Hence the Lagrange multipliers are always non-negative. Our optimization problem in the primal can be written as:

$$\min_w \max_\Lambda \mathcal{L}(w, \Lambda) \tag{5}$$

Instead of solving the primal in (5), we often solve its corresponding dual:

$$\max_\Lambda \min_w \mathcal{L}(w, \Lambda) \tag{6}$$

We make two comments on our formulation. First, the use of the hinge function achieves two objectives: (a) no penalty is paid when constraints are satisfied, and (b) the number of dual variables is reduced from $O(mK)$ to $O(K)$, making the formulation scalable. Second, our formulation can handle arbitrary constraints as long as they are differentiable. Note that even simple form of constraints (such as linear) over the output variables typically represent highly non-linear functions of the networks weights.

| Constraint: C | $g(w)$:**Choice 1** | $g(w)$: **Choice 2** |
|---|---|---|
| $y_j = v$ | $P_w(y_j = v)$ | |
| $\neg C_1$ | $1 - g_1(w)$ | |
| $C_1 \vee C_2$ | $\min(g_1(w) + g_2(w), 1)$ | $\max(g_1(w), g_2(w))$ |
| $C_1 \wedge C_2$ | $\max(g_1(w) + g_2(w) - 1, 0)$ | $\min(g_1(w), g_2(w))$ |

Table 1: $g(w)$, $g_1(w)$ and $g_2(w)$ are soft value functions for $C$, $C_1$ and $C_2$, respectively.

.

## 3.2 Constraint Language for Discrete Output Spaces

A common learning scenario for many problems is when each element of the target $y$ belongs' to a discrete space. In such cases, each $y$ is given as a vector $(y_1, y_2, \cdots, y_r)$, where each $y_j \in \mathcal{V}$ where $\mathcal{V}$ represents a set of discrete values. The network output is then represented as an $r$ dimensional vector, where each element of the vector represents a probability distribution over $\mathcal{V}$, i.e. $P_w(y_j|x) \ \forall j, 1 \leq j \leq r$. One of the common loss functions for discrete spaces is the cross entropy based loss though other loss functions can also be used. At prediction time, given a new test example $x$, we output the vector of values which have the highest probability for each element $y_j$ in the output space, i.e., $\text{argmax}_{y_j} P_w(y_j|x), \forall j, 1 \leq j \leq r$. In this section, we lay out the details of a language which can handle logical constraints specified over discrete output spaces as described above. Our formulation is based on soft logic used earlier in the literature Bröcheler *et al.* [2010], and represents constraints in the form of inequalities: $f_k(w) \leq 0$ where $w$ are network weights.

Our constraints are defined as logical expressions over values $v$ that each $y_j$ can take. A constraint $C$ can take the following form: (a) $C : \mathbb{1}\{y_j = v\}$ (b) $C = \neg C_1$ (c) $C = C_1 \vee C_2$ (d) $C = C_1 \wedge C_2$. Here, $C_1, C_2$ denote constraints constructed recursively using above rules. The first expression (a) can be thought of as an atomic constraint, and rest are constructed by applying logical operators over existing constraint(s). Note that $C_1 \rightarrow C_2$ can be written as $\neg C_1 \vee C_2$. Given a logical constraint $C$ over the values output by a network with parameters $w$, we construct a function $g(w) \in [0, 1]$ which denotes the soft value of the corresponding logical expression. Table 1 describes conversions from a logical expression to a corresponding (soft) value. Finally, given a constraint $C$ and the associated function $g(w)$, the corresponding constraint can be written as: $g(w) = 1$. Since $g(w) \in [0, 1]$, it is equivalent to: $g(w) \geq 1$ or $f(w) = 1 - g(w) \leq 0$. We note that since all our constraints are over variables with probability distributions defined over them, introducing soft logic does not make the constraints any softer, it only gives a way to combine underlying probability values.

## 4 Training

**Supervised:** We solve the dual optimization problem described in Equation 6 by alternating gradient descent (ascent) steps over $w$ and $\Lambda$, respectively. The gradients of the $\mathcal{L}$ with respect to $w$ and $\lambda_k$ are given as:

$$\nabla_w \mathcal{L}(w; \Lambda) = \nabla_w L(w) + \sum_{k=1}^{K} \lambda_k \nabla_w h_k(w); \quad \frac{\partial \mathcal{L}(w; \Lambda)}{\partial \lambda_k} = h_k(w), \forall k. \quad (7)$$

Non-differentiability due to the Hinge function in $h_k$ can be handled by using sub-gradients. Correspondingly, the parameter update equations can be written as:

$$w^{(t_1+1)} \leftarrow w^{(t_1)} - \alpha_w \nabla_w \mathcal{L}(w; \Lambda); \quad \Lambda^{(t+1)} \leftarrow \Lambda^{(t)} + \alpha_\Lambda \nabla_\Lambda \mathcal{L}(w; \Lambda) \quad (8)$$

Algorithm 1 presents the pseudocode for our learning algorithm. Initially, $w$ are updated for a *warmup* number of iterations with each $\lambda_k = 0$ (i.e., no constraints). Then, we perform the following in succession: for every one update of $\Lambda$ parameters, we update the $w$ parameters for $l$ steps, where $l$ grows based on an arithmetic progression in increments of $d$. Intuitively, this ensures that ratio of the effective learning rates for $w$ updates and $\Lambda$ updates goes to infinity with increasing number of $\Lambda$ updates as $l \rightarrow \infty$. For convergence, we resort to the theory of min-max optimization presented by Jin *et al.* [2019]. Their key result states that for a min-max optimization problem, alternating gradient ascent (descent) over max (min) variables converges to the local min-max point (analogous of local minima in the single variable case) if the ratio of learning rates of inner and outer variables goes to $\infty$ in the limit. A significant advantage of our formulation is that in practice the inner loop can often involve application of algorithms such as AdaDelta or RMSProp, which perform

gradient descent, but we may not have direct control over the learning rate for $w$ parameters. But our step based update still ensures that effective ratio of learning rates goes to infinity. We state this formally in our next theorem (see supplement for a proof).

**Theorem 1.** *Algorithm 1 converges to a Local minmax point of $\mathcal{L}(w; \Lambda)$ for any $d \geq 1$.*

**Semi-supervised:** Our framework can be easily extended to the case of semi-supervised learning. Since we do not have the target value $y$ for unlabeled examples, we can't compute the loss (cross-entropy term) in expression for $\mathcal{L}(w; \Lambda)$ in Equation 4 and hence, contribution of unlabeled examples to this term is ignored. On the other hand, the second term in the expression for $\mathcal{L}(w; \Lambda)$ (corresponding to constraints) does not depend on the target values $y$. Therefore, for unlabeled examples, we can take this contribution into account by computing this term just like in the case of labeled examples. As demonstrated by our experiments, this simple idea of using unlabeled data only for enforcing the constraints can act as a strong regularizer and result in significantly improved models, especially when there is small amount of labeled data available for training. This is also observed in earlier work (Xu *et al.* [2018]; Mehta *et al.* [2018]).

---

**Algorithm 1** Training of a Deep Net with Constraints. Hyperparameters: $warmup, d, \beta, \alpha_\Lambda^0, \alpha_w$

1  Initialize: $w$ randomly; $\lambda_k = 0, \ \forall k = 1 \ldots K$
2  **for** $warmup$ *iterations* **do**
3     |    **Update** $w$**:** Take an SGD step wrt $w$ on $\mathcal{L}(w; \Lambda)$ on a mini-batch
4  Initialize: $l = 1; t = 1; t_1 = 1; \alpha_\Lambda = \alpha_\Lambda^0$
5  **while** *not converged* **do**
6     |    **Update** $\Lambda$**:** Take an SGA step wrt $\Lambda$ on $\mathcal{L}(w; \Lambda)$ on a mini-batch
7     |    Increment $t = t + 1$
8     |    **for** $l$ *steps* **do**
9     |       |    **Update** $w$**:** Take an SGD step wrt $w$ on $\mathcal{L}(w; \Lambda)$ on a mini-batch
10    |       |    Increment $t_1 = t_1 + 1$
11    |    Update $l = l + d$
12    |    Set learning rates: $\alpha_\Lambda = \alpha_\Lambda^0 \frac{1}{1+\beta t}$

---

## 5   Experiments

The goal of our experiments is to answer three questions. (1) Does constrained training help in learning more accurate models, especially in the low data setting? (2) Does constrained training result in models with better constraint satisfaction at prediction time? (3) What is the impact of semi-supervision? We perform experiments on three different NLP benchmarks, which we describe next. The specific details of software environments and hyperparameters are mentioned in the supplement.

### 5.1   Semantic Role Labeling (SRL)

Given a sentence with a predicate (verb), the goal of SRL is to extract and label the arguments for it to determine who did what to whom, when and where, etc. In SRL literature, there is a long history of using linguistic and structural constraints in inference (e.g., Punyakanok *et al.* [2008]). We assess the value of constraints in learning more robust neural models.

**Dataset & Baseline Model:** We use English Ontonotes 5.0 dataset[1] using the CONLL 2011/12 shared task format (Pradhan *et al.* [2012]) as the training data. The labeling task is modeled as sequence labeling using the BIOUL encoding. The baseline model (B) uses a deep Bidirectional LSTM, initialized with ElMo+Glove embeddings.[2]

**Constraints:** We impose two types of constraints. (1) Syntactic Constraints: let $SY = \{(a,b)|a < b\}$ be the set of syntactic spans of a sentence in its syntactic parse tree. Let $y_j^{B_l}$ and $y_j^{L_l}$ be the indicator variables corresponding to beginning and end (last) tag of argument label $l$ at $j^{th}$ word. Then syntactic constraints can be written as $y_a^{B_l} \implies \bigvee\limits_{j \in \{b:(a,b) \in SY\}} y_j^{L_l}$ and $y_b^{L_l} \implies \bigvee\limits_{j \in \{a:(a,b) \in SY\}} y_j^{B_l}, \forall a, b, l$.
These constraints are similar to those used by Mehta *et al.* [2018], albeit in a different formulation.

| Scenario | F1 Score | | | Total Constraint Violations | | |
|---|---|---|---|---|---|---|
| | 1% Data | 5% Data | 10% Data | 1% Data | 5% Data | 10% Data |
| B | 62.99 | 72.64 | 76.04 | 14,857 | 9,708 | 7,704 |
| CL | 66.21 | 74.27 | 77.19 | 9,406 | 7,461 | 5,836 |
| B+CI | 67.90 | 75.96 | 78.63 | 5,737 | 4,247 | 3,654 |
| CL + CI | 68.71 | 76.51 | 78.72 | 5,039 | 3,963 | 3,476 |

Table 2: Effect of constrained learning on SRL, with and without constrained inference (CI).

(2) Transition Constraints: BIOUL encoding naturally defines valid transitions for a sequence, e.g., $L_l$ must be preceded by $B_l$ or $I_l$. For a given tag $t$, let $V_t$ be the set of valid tags for the next word. Then, transition constraints enforce that: $\forall\, j, t : y_j^t \implies \bigvee_{u \in V_t} y_{j+1}^u$.

**Methodology:** We compare against two different models, the baseline (B), and the baseline augmented with Viterbi decoding (B+CI). This constrained decoding enforces transition constraints at inference time. We name the constrained learning versions of these algorithms by CL and CL+CI, respectively. Note that for test instances, the syntactic spans $SY$ are not available. We use the standard train/dev/test split and use the official Perl script to compute span based F1-scores. We train with 1%, 5% and 10% of training data selected randomly.

**Results:** Table 2 presents our results. We observe significant F1 gains of constrained learning (B+CL) over the baseline B, supporting the hypothesis that constraints can help in learning more robust models. We find that constrained learning with constrained decoding consistently performs the best, even though marginal improvements over B+CI are smaller.[3] We also note that the benefit of constrained learning decreases as training data increases, suggesting that this approach is most useful in low data settings. In addition to F1-scores, we also report total number of constraint violations and find that constrained learning consistently makes significantly fewer violations. At the same time, we note that there are still substantial violations remaining. This is not entirely suprising, since learning span constraints without known spans is akin to learning a significant aspect of the syntactic parsing task, making the learning task much harder.

## 5.2 Named Entity Recognition (NER)

The task corresponds to assigning a tag for each word from a given set of NER tags, e.g., 'location', 'person' etc. In addition, we also assume that the (training) dataset is labeled with Part of Speech (POS) tags for each word. This information is readily available for many datasets. We treat POS tagging as an auxiliary task in the standard multi-task learning (MTL) framework.

**Dataset & Baseline Model:** We use the publicly available GMB[4] dataset (Bos *et al.* [2017]) in our experiments. It contains about 62 thousand sentences, 24 different NER tags and 43 different POS tags. We randomly split it into 60/20/20 train/dev/test sets respectively. After removing the hierarchy among NER tags (e.g., mapping 'person-title' and 'person-family-name' to a single 'person'), we are left with 9 high-level NER tags. We use the BIO encoding in our modeling. Our baseline model (B) is a BiLSTM that is setup in an MTL framework for predicting both NER and POS. For both the tasks, we use a single BiLSTM layer whose parameters are shared between the two tasks.

**Constraints:** We encode our prior linguistic knowledge about the relationships between NER and POS as constraints – for any NER tag $t_e$, we have an allowed set of POS tags $T_p(t_e)$. If a word takes an NER tag $t_e$, then its POS tag must come from the set $T_p(t_e)$, i.e., $y_j^{NER} = t_e \Rightarrow y_j^{POS} \in T_p(t_e)$. Here, $y_j^{NER}$, $y_j^{POS}$ are the output variables corresponding to NER and POS tags for the $j^{th}$ word, respectively. We give the full details of our constraints in the supplement.

**Methodology:** We compare the following models. (1) B: Baseline, (2) CI: Constrained Inference, (3) CL: Constrained Learning, and (4) SCL: Semi-supervised Constrained Learning. B is the base model, CI does regular training with constrained inference using dual decomposition (Rush and Collins [2012]), CL is our model doing constrained training (supervised) and SCL does constrained training using semi-supervised data. In order to test the performance of our model in low data setting, we randomly select data subsets of sizes {400, 800, 1600, 6400, 12800, 25600, 37206} and use them for

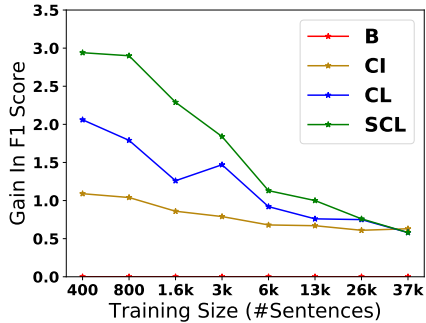

(a) Avg. Gain in F1 Score Over Baseline.

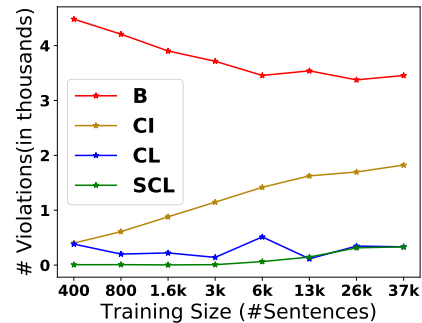

(b) Avg. number of Constrained Violations

Figure 1: NER: Comparison of different training techniques. B: Baseline; CL: Constrained Learning; SCL: Semi-supervised Constrained Learning; CI: Constrained Inference

training each model. In each case, the data not used for training is used as unlabeled data for SCL (after removing the labels). The reported results are averaged over 10 different randomly selected samples for each training size.

**Results:** Figure 1a compares the performance of the four models. We plot the baseline model at zero, and plot the performance of all other models relative to the baseline (see supplement for absolute numbers and standard deviations). There is a good gain in F1-score when learning with constraints, with most gain obtained for smaller training sizes. Semi-supervision results in significant additional gains. Figure 1b plots the number of constraint violations with varying training size. For CL and SCL, this number is close to 0 all through. Counter intuitively, the violations increase monotonically for CI. This is because with less training data, learning is very shallow, resulting in 'Other' prediction most of the time, and the constraints are trivially satisfied. As learned model becomes more complex, CI finds it harder to satisfy the constraints without hurting the performance. We do early stopping of dual decomposition based on dev set performance. This results in decent F1 but high constraint violations. If run till convergence, CI results in all constraints being satisfied but with performance lower than the baseline. CL does not suffer from this phenomenon due to constraint aware learning.

We also experiment with using CI (constrained inference) on top of CL and SCL; this results in no additional gains, since most constraints are already being satisfied. This highlights that constrained learning may sometimes obviate the need for constrained inference, This can lead to a huge reduction in precious test times. For instance CI has test times 3-15 times that of CL, depending on testing batch size.

### 5.3 Fine Grained Entity Typing

This is a multi-label classification problem. Given a set $M$ of textual mentions of an entity $e$, we are interested in finding all the types that the mentions in $M$ belong to (Yao *et al.* [2013]; Verga *et al.* [2017]; Murty *et al.* [2018]). Note that the labels here are entity types.

**Dataset & Baseline Model:** We work with Typenet[5] (Murty *et al.* [2017]), a publicly available dataset of hierarchical entity types for extremely fine-grained entity typing. It has been curated by mapping Freebase (Bollacker *et al.* [2008]) types into Wordnet (Miller [1995]) hierarchy. The dataset contains over 1,900 types, placed in a hierarchy of average depth of 7.8. It also provides a corpus of textual mentions extracted from Wikipedia articles. It contains 344,246 entities mapped to 1,081 types arranged in the type hierarchy. For baseline (B), we use the state of the art model proposed for this task by Murty *et al.* [2018].[6] Each mention $m$ is represented by an encoding computed using a CNN over the sentence, and each type is represented using an embedding vector. The two are combined to get a similarity score. Scores coming from different mentions in a set are pooled to get a final score for each entity. To exploit the hierarchical structure, an additional loss term (H) encourages entities close in the hierarchy to get similar embedding vectors. This can be thought of as imposing a soft constraint on the entity types. We compare with both these versions in our experiments.

| | MAP Scores | | | Constraint Violations | | |
|---|---|---|---|---|---|---|
| Scenario | 5% Data | 10% Data | 100% Data | 5% Data | 10% Data | 100% Data |
| **B** | 68.62 | 69.21 | 70.47 | 22,715 | 21,451 | 22,359 |
| **B+H** | 68.71 | 69.31 | 71.77 | 22,928 | 21,157 | 24,650 |
| **CL** | 80.13 | 81.36 | 82.80 | 25 | 45 | 12 |
| **SCL** | 82.22 | 83.81 | | 41 | 26 | |

Table 3: TypeNet: MAP Scores (in %) and # of constraint violations for different training sizes

**Constraints:** We enforce two types of constraints in our model. (a) Type Inclusion: given two types $t_i$ and $t_j$ such that $t_i$ is ancestor of $t_j$ in the type hierarchy, for any entity, if $t_i$ is selected as a possible entity type, then $t_j$ should also be selected. I.e., $y_{t_i} \Rightarrow y_{t_j}$ where $y_{t_i}$ and $y_{t_j}$ are indicators variables for the corresponding types being selected. This results in 1,891 constraints. (b) Type Exclusion: pairs of types $t_i$ and $t_j$ (e.g., 'library' and 'camera') that should not co-occur for any entity. I.e., $y_{t_i} \Rightarrow \neg y_{t_j}$. This results in a total of about 555,000 constraints.

**Methodology:** We compare four different models: (a) B: Baseline, (b) B+H: Baseline with hierarchically constrained embeddings (c) CL: constrained learning (d) SCL: constrained learning with semi-supervision. Our constrained learning models are learned on top of vanilla baseline (and do not make use of hierarchical embeddings). We use the original splits of 90%, 5% and 5% for training, validation and testing, respectively (Murty *et al.* [2018]). We compare the performance of the four models for training at (1) 5% of the data (2) 10% of the data, and (3) full training set. The smaller training subsets are chosen randomly. As earlier, any unused data in the training fold is used for semi-supervision (after removing labels).

**Results:** Table 3 presents our comparison results. We note that our baseline results are significantly higher than those reported in Murty *et al.* [2018]. We believe this is because they did not train the model until convergence; running till convergence results in significantly higher numbers. After additional training, the relative advantage of B+H model over B as reported in their paper is lost. Our constrained model (CL) can give up to 11 pt increase in the performance both at 5% of the data as well as 10% of the data. With semi-supervision, this gain hovers in the range of 12-14 pts. There is three orders of magnitude drop in the number of constraint violations when using constrained learning. Interestingly, CL performance with 100% data is slightly worse than semi-supervision with 10% data. We hypothesize that the reason is noise in training data in terms of either missing or incorrect labels. We note that there are $634,544$ type inclusion constraint violations in the training data containing $294,781$ entities. As a result, more noisy data is likely hurting the performance.

We also compare our constrained learning against Diligenti *et al.* [2017a]'s approach of using soft constraints, where the violation penalty is multiplied with a constant $\lambda$ and added to the original loss. When using the best values of the $\lambda$ parameter, we find that both methods perform similarly. However, Diligenti's performance requires extensive search over the $\lambda$ parameters, and varies significantly based on the value of $\lambda$. On the other hand, our formulation can implicitly discover optimal $\lambda_k$ values (we have one for every constraint) by way of our Langrangian formulation. This obviates the need for an explicit search for $\lambda$ which can be expensive. In fact, setting constant $\lambda$ to be the average of $\lambda_k$'s from our formulation gives its best score.

## 6  Conclusion and Future Work

In this paper, we have proposed a primal-dual based approach for solving the problem of learning with hard constraints in deep learning models. While earlier work has modeled the constraints as soft, incorporating penalty in the loss term, we instead directly optimize the hard constraints using a Langrangian based formulation. We show that our algorithm converges to local min-max points of the objective. For the case of discrete output spaces, we also present a constraint language using soft logic. Experiments on three different NLP tasks show the effectiveness of our approach compared to non-constrained baselines, as well as constrained inference, achieving the state-of-the-art results in two of the domains. In one of the domains, our approach completely eliminates the need for expensive constrained inference. Directions for future work include learning constraints automatically, and experimenting on non-NLP tasks. We have made our all our code publicly available at: *https://github.com/dair-iitd/dl-with-constraints* for future research.

## Acknowledgements

We thank IIT Delhi HPC facility[7] for computational resources, which allows us to run experiments at large scale. We thank Guy Van den Broeck, Yitao Liang, Sanket Mehta, Shikhar Murty, Dan Roth, Alexander Rush and Vivek Srikumar for useful discussions on the work. We also thank Deepanshu Jindal for proofreading our code. Mausam is supported by grants from Google, Bloomberg and 1MG. Parag Singla is supported by the DARPA Explainable Artificial Intelligence (XAI) Program with number N66001-17-2-4032. Both Mausam and Parag Singla are supported by the Visvesvaraya Young Faculty Fellowships by Govt. of India and IBM SUR awards. Any opinions, findings, conclusions or recommendations expressed in this paper are those of the authors and do not necessarily reflect the views or official policies, either expressed or implied, of the funding agencies.

## Footnotes

[1] *http://cemantix.org/data/ontonotes.html*

[2] implemented in *https://allennlp.org/models#semantic-role-labeling*

[3]Our F1-scores are not directly comparable with those reported in Mehta *et al.* [2018], since their exact training splits (or code) are unavailable. Overall, our gains due to constrained learning are similar to theirs.

[4]*https://gmb.let.rug.nl/data.php*

[5] *https://github.com/iesl/TypeNet*

[6] https://github.com/MurtyShikhar/Hierarchical-Typing

[7]*http://supercomputing.iitd.ac.in*

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
