[Supplementary Material · Supplement Neurips_2019_Primal_Dual_Formulation_for_Deep_Learning_with_Constraints.pdf]

# Supplementary Material: Primal-Dual Formulation for Deep Learning with Constraints

**Yatin Nandwani, Abhishek Pathak, Mausam and Parag Singla**
Department of Computer Science and Engineering
Indian Institute of Technology Delhi
{yatin.nandwani,abhishek.pathak.cs115,mausam,parags}@cse.iitd.ac.in

## 4 Training

---

**Algorithm 1** Training of a Deep Net with Constraints. Hyperparameters: $warmup, d, \beta, \alpha_\Lambda^0, \alpha_w$

---

1  Initialize: $w$ randomly; $\lambda_k = 0, \ \forall k = 1 \ldots K$
2  **for** $warmup$ $iterations$ **do**
3  $\quad$ **Update** $w$**:** Take an SGD step wrt $w$ on $\mathcal{L}(w; \Lambda)$ on a mini-batch
   **end**
4  Initialize: $l = 1; t = 1; t_1 = 1; \alpha_\Lambda = \alpha_\Lambda^0$
5  **while** $not\ converged$ **do**
6  $\quad$ **Update** $\Lambda$**:** Take an SGA step wrt $\Lambda$ on $\mathcal{L}(w; \Lambda)$ on a mini-batch
7  $\quad$ Increment $t = t + 1$
8  $\quad$ **for** $l$ $steps$ **do**
9  $\quad\quad$ **Update** $w$**:** Take an SGD step wrt $w$ on $\mathcal{L}(w; \Lambda)$ on a mini-batch
10 $\quad\quad$ Increment $t_1 = t_1 + 1$
   $\quad$**end**
11 $\quad$ Update $l = l + d$
12 $\quad$ Set learning rates: $\alpha_\Lambda = \alpha_\Lambda^0 \frac{1}{1+\beta t}$
   **end**

---

**Theorem 1.** *Algorithm 1 converges to a Local minmax point of $\mathcal{L}(w; \Lambda)$ for any $d \geq 1$.*

*Proof.* Without loss of generality, we can assume $warmup = 0$. Then, for a given $t$ (number of $\Lambda$ updates), let $t_1$ denote the number of corresponding $w$ updates. Then, $t_1 = 1 + d + \cdots + t * d$, i.e., $t_1 = O(t^2 d)$. Therefore, the ratio of effective learning rates for $w$ and $\Lambda$ updates $= \frac{\alpha_w}{\alpha_{\Lambda_0}}(1 + \beta t)O(td)$. This term goes to $\infty$ with increasing $t$. Hence, by Theorem 28 in Jin *et al.* [2019], Algorithm 1 converges to the local Minmax point of $\mathcal{L}(w; \Lambda)$. $\square$

## 5 Experiments

**Optimizer:** For $w$ updates, we use the same optimizer as used in the base model and for $\lambda$ updates we use SGD with momentum of $0.9$.

**Software Used:** All models are trained using PyTorch [1]. For NER and SRL experiments, we use Allennlp [2] library which is built on top of PyTorch.

**Computational Resources:** All our models are trained on PADUM: Hybrid High Performance Computing Facility at IITD [3].

## 5.1 Semantic Role Labeling

**Hyperparameters:** In all the experiments, $warmup$ iterations and initial value of $l = l_0$ is selected in the same way: to select $warmup$ iterations, we train the base model without constraints till convergence and as a rule of thumb, select $warmup$ iterations as the iteration number where it reaches around 25% of its peak performance. Initial value of $l = l_0$ is set arbitrarily at 10. Initial learning rate $\alpha_\Lambda^0$ is fixed at 0.05. Constant $d$ and learning rate decay parameters $\beta$ is selected through a grid search over $\{1, 10\}$ and $\{1, 1/5, 1/10\}$ respectively and we select the best combination based on the performance over dev set. Table 1 enumerates the best value of these two hyper-parameters for different training sizes.

|  | Constant $d$ | Decay $\beta$ |
|---|---|---|
| **1% Data** | 1 | 1/5 |
| **5% Data** | 10 | 1 |
| **10% Data** | 10 | 1 |

Table 1: Best hyper-parameters in SRL experiments for different training sizes.

## 5.2 Named Entity Recognition

**Constraints:** Below we enumerate the constraints that we impose on the NER and POS label for any given word.

B-org $\implies$ {NNP}
B-tim $\implies$ {NNP, CD, JJ}
B-geo $\implies$ {NNP}
B-gpe $\implies$ {JJ, NNS, NNP}
B-per $\implies$ {NNP}
B-eve $\implies$ {NNP}
B-art $\implies$ {NNP, NNPS, JJ, NNS}
B-nat $\implies$ {NNP}
I-per $\implies$ {NNP}
I-org $\implies$ {NNP}
I-geo $\implies$ {NNP, NNPS}
I-tim $\implies$ {CD, NNP, NN, IN}
I-eve $\implies$ {NNP}
I-art $\implies$ {NNP}
I-gpe $\implies$ {NNP}
I-nat $\implies$ {NNP}

**Hyperparameters:** $warmup$ iterations and initial value of $l$ are selected as in SRL experiments. In these experiments, we do not decay the learning rate and set $\beta$ to 0. To select the learning rate $\alpha_\Lambda$, and constant $d$, we do a grid search over $\{0.01, 0.05\}$ and $\{1, 5\}$ respectively and select the best combination based on the performance over dev set. Table 2 enumerates the best value of these two hyper-parameters for different training sizes in both the settings: constrained learning and semi-supervision.

**Results** Table 3 enumerates the mean F1-Score over 10 random shuffles of data, along with its stdev, for different models with varying training size. We also tabulate the number of violations in each scenario.

|  | CL | | SCL | |
|---|---|---|---|---|
| Training Size | Learning Rate $\alpha_\Lambda$ | Constant $d$ | Learning Rate $\alpha_\Lambda$ | Constant $d$ |
| 400 | 0.05 | 5 | 0.01 | 5 |
| 800 | 0.05 | 1 | 0.01 | 5 |
| 1,600 | 0.05 | 1 | 0.05 | 1 |
| 3,200 | 0.05 | 1 | 0.05 | 1 |
| 6,400 | 0.01 | 5 | 0.01 | 5 |
| 12,800 | 0.05 | 1 | 0.01 | 5 |
| 25,600 | 0.01 | 5 | 0.01 | 5 |
| 37,206 | 0.01 | 5 | 0.01 | 5 |

Table 2: Best hyper-parameters in NER for different training sizes in both scenarios: CL and SCL.

| Train Size | F1-Score(Mean $\pm$ Stdev) | | | | Mean #Violations | | | |
|---|---|---|---|---|---|---|---|---|
| | B | CL | SL | CI | B | CL | SL | CI |
| 400 | $51.6 \pm 0.99$ | $53.7 \pm 1.16$ | $54.6 \pm 0.83$ | $52.7 \pm 0.79$ | 4,482 | 383 | 7 | 401 |
| 800 | $57.3 \pm 1.45$ | $59.1 \pm 1.34$ | $60.2 \pm 0.74$ | $58.3 \pm 1.25$ | 4,208 | 201 | 8 | 610 |
| 1,600 | $62.3 \pm 1.05$ | $63.6 \pm 0.51$ | $64.6 \pm 0.71$ | $63.2 \pm 0.84$ | 3,902 | 222 | 4 | 880 |
| 3,200 | $66.2 \pm 0.59$ | $67.7 \pm 0.38$ | $68.1 \pm 0.5$ | $67 \pm 0.55$ | 3,715 | 141 | 8 | 1,147 |
| 6,400 | $69.8 \pm 0.54$ | $70.8 \pm 0.34$ | $71 \pm 0.43$ | $70.5 \pm 0.53$ | 3,456 | 514 | 64 | 1,418 |
| 12,800 | $72.1 \pm 0.28$ | $72.9 \pm 0.36$ | $73.1 \pm 0.38$ | $72.8 \pm 0.27$ | 3,540 | 115 | 147 | 1,626 |
| 25,600 | $74.3 \pm 0.24$ | $75.1 \pm 0.17$ | $75.1 \pm 0.25$ | $74.9 \pm 0.2$ | 3,376 | 347 | 315 | 1,697 |
| 37,206 | $75.3 \pm 0.24$ | $75.8 \pm 0.21$ | $75.8 \pm 0.21$ | $75.9 \pm 0.25$ | 3,455 | 333 | 333 | 1,823 |

Table 3: F score for different models (mean $\pm$ stdev), along with average number of constraint violations

## 5.3 Fine Grained Entity Typing

$warmup$ iterations and initial value of $l$ are selected as in the above two experiments. As in NER experiments, we do not decay the learning rate and set $\beta$ to 0. We observed that higher values of the constant $d$ hurts the performance and increase the number of constraint violations as well. Hence, we set it to 0 which gives the best results. To select the learning rate $\alpha_\Lambda$, we do a grid search over $\{0.01, 0.02, 0.03, 0.04, 0.05\}$ and select the best value based on the performance over dev set. Table 4 below enumerates its best value different training sizes in both the settings: constrained learning and semi-supervision.

|  | CL | SCL |
|---|---|---|
| Training Size | Learning Rate $\alpha_\Lambda$ | Learning Rate $\alpha_\Lambda$ |
| 5% Data | 0.05 | 0.01 |
| 10% Data | 0.02 | 0.03 |
| 100% Data | 0.04 | |

Table 4: Best hyper-parameters in Typenet for different training sizes in both scenarios: CL and SCL.

## Footnotes

[1]https://pytorch.org/

[2]https://allennlp.org/

[3]*http://supercomputing.iitd.ac.in*

# References

Chi Jin, Praneeth Netrapalli, and Michael I. Jordan. Minmax optimization: Stable limit points of gradient descent ascent are locally optimal. *CoRR*, abs/1902.00618, 2019.