[Reviews · NeurIPS 2019]

Reviewer 1



The paper proposes using lagrange multipliers + stochastic gradient descent to incorporate constraints during the learning of deep neural networks. Both the incorporation of lagrange multipliers (except for the nice hinge trick to turn N constraints into 1 constraint and limit the number of dual variables) and the language to encode constraints are relatively straightforward and easy to follow. The results look positive, and seem to show an improvement over applying constraints only at inference time, and a nice gain from using the constraints in a semi supervised setting; both of which reproduce constrained learning results which were known in linear models but haven't been replicated yet in deep models. ----- after author response No changes to my score.

Reviewer 2



(After reading the author response) My concerns about the experiments, mismatching table vs. figures and why the reported numbers are very different from the previous work, are well resolved. This was the main reason for my doubt on this work, and as it is resolved I change my position to accepting this paper. Especially I liked the explanation on [sig&qual] #2. But, I don’t think I read this in the main text and It would be interesting to include this to the main text or supplementary material. Lastly, on the originality & contextualizing to the previous work, the main point of criticism was to better contextualize the relation to the previous work. I don’t think regarding previous work as task-specific formulation just because they ‘experimented’ on a single task is not the right way. (Maybe it is better to say that they only verified on the specific task rather than regarding their method as task-specific.) In my opinion, this applies to both Diligenti & Metha. For example, the reason I brought up Lee et al 2017 was not that I wasn’t aware that this is the work on inference. However, Mehta et al 2018 use the same constraint loss as Lee et al 2017 and latter work experiments on three different settings besides SRL. Combining these two facts, it is not hard to realize that the formulation of Mehta et al 2018 is not really task-specific. The major difference between Diligenti et al. and authors’ work, I think is, that the author is using ‘full Lagrangian formulation’ as mentioned in rebuttal, and this work is doing that. I think the same difference applies to Mehta et al. as well. (Fixed weight regularization vs. min-max optimization) albeit there is more difference in the formulation of constraint as well. So why don’t authors highlight this difference more instead of regarding previous work as ‘task-specific’? This difference has better benefits as well, as noted in rebuttal, (a) the performance of [sig&qual] #2, (b) benefit over expensive grid search. If the paper gets accepted, I wish the authors to reflect these concerns. ---------------------------------------------------------------------------------------------- (Initial review) [Summary ] The paper presents a training algorithm for a deep network with hard constraints on output labels. The paper converts the constrained optimization problem to min-max optimization using Lagrangian function. To show the efficacy of the model, three experiments are conducted in 5.1 SRL, 5.2 NER and 5.3 Fine grained entity typing. The paper brings in a structured way of training with output constraints. However, I am not sure how much gain this model has on top of fixed weight on constraints (Metha et al 2018 & Diligenti et al 2017) with the provided experiments. Also while the experiments seem convincing as itself, it is hard to see how much significance this work brings in as the baselines significantly differ with related work. Also, it would give a better picture of this method if the paper could provide more analysis: an analysis on convergence, an analysis on experiment results on why more labeled data sometimes hurt, etc. [originality ] 1. The full Lagrangian expression and linking the output constraint to the model parameter and optimizing them with subgradient seems novel. However, how exactly the authors formulate f(w) is unclear to me. Is it just following the way Diligenti 2017 does it? 2. On this line, is the difference with Diligenti 2017 mainly on min-max alternation (max part of \lambda)? If authors could clarify these points, it would be easier to evaluate the originality of this work. 3. Asserting convergence guarantee by referencing Jin et al. 2019 seems to be nice part of this work. But what is your criteria for convergence? How long does it take to converge generally? How much change does dual variables have (do important constraint get more weight in the end? If there is such notion of more important constraint) 4. I think the authors have good background knowledge. But, positioning this paper as general and other previous work (Diligenti et al. 2017 and Mehta et al. 2018) as task-specific is correct. As this papers using exact constraint language that Diligenti et al. 2017 provided, I thought the methodology was not task-specific. The same applies to Mehta et al., 2018. They simply use normalized constraint violation score which seems to be general enough. ( In fact, Lee et al., 2017 (by the same authors) show more than on applications beside SRL as well.) —> Also, please update the citation of Lee et al. 2017 citation to the conference paper (AAAI19). 5 Regarding the statement on (212) that this paper is the first to employ min-max optimization on neural network—> consider citing these two papers: Chen et al. ,A Primal-Dual Method for Training RNN Constrained by the Echo-State property, 2013 & Zhang et al., Primal–Dual Neural Networks, 2013. I did not read the 2nd paper, but the first one does the alternating min-max optimization albeit not for the constraint on the output space. [ clarity ] 1. While the paper is well written (well motivated with good coverage of related work) there is certainly room for improvements in method section (sec3) and on the description of the constraints. 2. I think the authors show 3.3 in order to show how they can link constraint on the output to parameter w: C_k(\hat y) —> C_k(w) —> f_k(w) <0 but there is no text making the relation between 3.2 and 3.3. And 193-194 is confusing. f(w)>=1 (193) and f \in [0,1] (194) ?? As this is an important part of the paper, authors should take time to make the description clearer. 3. It was hard to understand what exact constraint of 5.1 and 5.2. Is constraint on 5.2 linguistically important? Perhaps give some examples to understand? Also in 5.1 it was hard to understand without referring to Mehta 2018. Lastly, isn’t (242) —> k \in {I:(I,j)\in SY} for the first condition. 4. There is a mismatch between Figure 1(a) and Table 1 in Appendix 5.1. SCL hurts the system and still, SCL is above the B (baseline) in figure 1(a). Could you explain this? 5. On NER task, the 24 tags were reduced to 9 tags for some reason, what was it? 6. Why does \lambda_k have to be positive in (5) Lagrange multiplier? As h_k(w)=0 is an equality constraint? 7. It would be good if there would be some explanation about the assumption in Jin et al. 2019’s proof and whether this work actually falls into it. 8. Typo: (225) (a) does constrained ‘trained' —> ‘training’ // need to close bracket on Table1: last row, choice 1 column. [significance & quality] 1. While the experiments look nice as presented, I wasn’t sure why the method has mismatches with other papers: Metha et al. 2018 for 5.1 and Murty et al. 2018 for 5.3 1. For 5.1, what is the reason for not comparing with Meta et al. when the application and constraint are exactly the same and the experiment setting and types of statistics (1%,10% F1 and degree of the constraint violation ) reported are very similar? Also, why are the performances much lower than that of Metha et al.(1%: 67.28—> 68.86, 10%: 78.56—> 79.34)? 2. (fine-grained entity typing.)I think there is a significant mismatch with the result of Murty et al. 2018. While your 5% data has 10 points higher MAP score than Murty 2018 the 100% dataset with hierarchy is much lower (6 points). 3. (fine-grained entity typing.) Also, the transitive feature-model (78.59% MAP score) of Murty 2018 seems to be comparable with '(a) type inclusion constraint' in 5.3 which I think these comparisons will be helpful for the readers. (5 point margin is still large) 2. It would be interesting how much gain the paper gets when compared with using a fixed \lambda. (Perhaps the same value for every constraint) This would give a good comparison point to the model proposed by Diligenti 2017. Along with this result, if there is an analysis on how much time (computation) difference the two approaches has, it will give a good cost-benefit tradeoff. [Questions on experiment results] 1. In Table 1 appendix 5.1, The SCL does not seem to hurt the performance as dataset becomes large (last two rows). Any remedies for these symptoms? And any explanation of why? Are the constraints not crucial for the task? The baseline has far more constraint violation but seems to perform stable and better. 2. In Table 1 of appendix 5.1, what is your intuition on CI performing better as the data size gets bigger even when the constraint satisfaction is less (constraint violation of CI > CL, SCL)? 3. For NER, is the combination of SCL + CI not showing any gains just byproduct of choice of CI? Meta et al. 2017. shows gains in similar settings on SRL. Have you tried the Lee et al.’s method as CI? 4. [Table3] What is your intuition for lesser constraint violation on for less labeled data on SCL?

Reviewer 3



I have read the author feedback and my evaluation remains the same. ============================================================ As I mentioned above, the choice of Lagrangian formulation seems natural (though the use in NLP problems could be still be novel and significant). The author cited Jin et al 2019 for the convergence of the optimizer. For better clarity, it would be good to discuss the specific conditions and conclusion of that result, and verify that the problems at hand do satisfy the conditions. Also, are (6) and (7) equivalent in this case (strong duality holds)? How to interpret figure 1, where CI is the best performer on the 37k training set in plot (a), whereas it still violates much of the constraints in plot (b)? This paper is quite NLP. Can authors think of additional application of the proposed approach in other domains (like in speech and vision?)

[Author Response · NeurIPS 2019]

Thank you for your detailed comments. We will make all clarifications below in the next version.

**R1:** • We have violations after CI since we do early stopping – satisfying them till end can sometimes hurt overall
performance since the model is not perfect. • We believe our novelty lies in the successful application of existing
constraint optimization ideas, mostly from non-neural world, in learning state-of-the-art Deep Learning models. While
general approach may be similar, details differ substantially. We note that our formulation in Sec 3.2 can handle any
kind of complex constraints, not just linear. Further, though constraints in our experiments are linear in output variables,
they are highly non-linear in the weight space. Learning the constraints automatically is a direction for future work.

**R2: [originality] #1,2:** There are important differences compared to Diligenti et al. (a) Unlike us, they do not work
with full Lagrangian formulation. Their approach is simply modifying the loss, and can not handle hard constraints; also
see the discussion in [sig&qual#2]. (b) Our formulation in Sec 3.2 is generic, and does not assume *any* specific form of
$f(w)$. It can work with arbitrary (including non-linear) constraints over the output variables, e,g. constraints over real
valued variables. There, the use of Hinge function is critical to ensure no penalty is paid when the constraints are not
violated, and also reduce the total number of dual variables. This is in contrast with Diligenti, which requires a specific
form for the constraint function: i.e. $0 \le f(w) \le 1$. (c) While it is true that our conversion of Boolean constraints to
soft-logic (Sec 3.3) is similar to Diligenti's, we note that both are based on prior work (Brocheler et al. UAI-10).

**#4:** We said that Diligenti and Mehta are task specific since they only experiment on a single task. Will clarify these
points. Lee at al. 2017 (and the AAAI-19 version) deal only with the constrained inference, not learning. We will
cite the two additional references (thanks!). **#3:** Jin et al. assume arbitrary non-convex non-concave form for a twice
differentiable function over which min-max has to be performed, and hence, it is also applicable to our formulation. Our
criteria for convergence (Algorithm 1) is when the change in both $w$ and $\Lambda$ parameters is less than some (respective)
thresholds. We missed a slight detail: we increase $l$ over successive $\Lambda$ updates using an AP. Then, in the limit $l \to \infty$
and using a slight variation of Theorem 28 in Jin et al., we are guaranteed to converge to a local min-max point (Defn.
14 in Jin et al.) of Lagrangian up to inclusion of some degenerate points. We will add this important theorem in the paper.
In practice, our algorithm converges very fast – it takes less than 2x time of training without constraints. This is not too
much cost to pay for doing hard constraint optimization and also getting significantly better results. • We see a large
variation in $\lambda$ values depending on the experiment, e.g., for NER they vary from 0.02 to 3.2. $\lambda$ for a constraint depends
on its degree of violation and not necessarily on its human perceived importance. Our algorithm is also significantly
faster than doing a grid search over a single $\lambda$ hyperparameter, which in addition to being restrictive, will also be much
more expensive – each training run would be as costly as the no-constraint training (also see [sig&qual#2]).

**[clarity] #2:** $f_k(w)$s used in 3.2 for the case of Boolean constraints is same as $f(w)$s in 3.3. $f(w) \ge 1$ ensures that
$f(w)$ stays equal to 1 (since we wanted to write inequality constraint). **#3:** Yes, all our constraints are linguistically
important. Will add examples in appendix. The first condition in line 242 is correct since the span should end (and not
start) at index $k$. **#4:** see response to [experiments #1] **#5:** We reduced the number of NER tags to 9 since we wanted to
work with 'higher level' NER tags (see line 262). **#6:** $\lambda_k$ is positive since each $h_k(w) \ge 0$ by construction, hence, the
equality $h_k(w) = 0$ is equivalently the inequality $h_k(w) \le 0$.

**[sig&qual] #1.1:** The goal of our expts was different from Mehta's. We were interested in examining whether the
model can learn effectively in presence of constraints. So we performed our expts *without* additional Viterbi decoding
at the end. Mehta used Viterbi in all expts – this explains difference in baselines. Nevertheless, based on your feedback,
we ran our expts with Viterbi decoding and could replicate their baseline (67.28%). We obtained 69.11 using CL
(supervised), which is 1.09 pt higher than their reported 68.02 for CL. Interestingly, our supervised performance is
0.25 pt higher than their semi-supervised learning (68.86)! Further, our CL model violates 14.36% of the syntactic
constraints as opposed to 20.49% and 19.38% for their constrained supervised and semi-supervised models, respectively.
We will report comparison for 10% data setting in final version. **#1.2:** The reason for higher performance than Murty et
al on 5% data is already explained in lines 323-325. Our B+H number for 100% data is lower than theirs due to a bug in
their code. Fixing this bug resulted in somewhat lower performance than reported in their paper. Nevertheless, even if
we take the number reported by them (75), our CL achieves 83.5 which is about 8 pts higher. **#1.3:** SCL performs 5 pts
higher than B+H+T in Murty et al. Will add this comparison. **#2:** We performed experiments using a fixed $\lambda$ (decided
using grid search) in Sec 5.3 for 5% data. It could only achieve a performance of 72.3 (vs. 78.4 for CL). This clearly
demonstrates that fixed $\lambda$ may not be enough for performance, in addition to requiring expensive grid search.

**[experiments] #1,2:** There was a typo in Table 1 (supplement) - last entry of SCL column for F1-score should be 75.66
(instead of 74.88). This makes it consistent with Fig 1(a) in the main paper and also answers the concern about SCL
doing worse than CL (which is not the case). The performance difference of various algorithms at 26k and 37k in
Figure 1(a) is not stat. significant. **#3:** We worked with dual decomposition as our CI for NER. We could not get Lee
et al.'s code despite repeated requests. **#4:** Violations are less on smaller amount of data for SCL, since most labels are
predicted as 'others' and it is easy to satisfy constraints.

**R3:** • see R2 response [originality]#3. • Strong duality doesn't hold since original objective and constraints can be
highly non-convex (in $w$). • The performance difference in Fig 1(a) at 26k and 37k is not stat. significant; it is quite
plausible for two algorithms to have same/similar performance, while one satisfies more constraints than the other. •
Our method is not just for NLP – exploring other application domains is a direction for future work.

[Meta-Review · NeurIPS 2019]

All reviewers were positive about the contributions in the paper so I recommend acceptance. Congratulations! Please take into account all the reviewers' comments when preparing the final version of the paper.